# Bat Dicer antiviral role and subcellular localization differ upon alphavirus infection in two distinct species

**Léa Gaucherand**[☯], **Hugo Marie**[☯], **Julie Cremaschi, Sébastien Pfeffer**[ID]*

Université de Strasbourg, Architecture et Réactivité de l'ARN, Institut de Biologie Moléculaire et Cellulaire du CNRS, Strasbourg, France

☯ These authors contributed equally to this work.

* s.pfeffer@ibmc-cnrs.unistra.fr

## Abstract

Bats are reservoirs for many viruses that frequently cause epidemics in humans and animals. It is thus critical to better understand their immune system and mechanisms of antiviral immunity. Despite an increasing number of studies, much remains to be discovered about the molecular mechanisms that govern bat-virus interactions, especially given the large diversity of bat species. Dicer is a conserved ribonuclease with multiple activities that can modulate antiviral immunity, including the detection of viral RNA as part of the RNA interference (RNAi) pathway, the maturation of microRNAs, and the direct inhibition of innate immunity in mouse and human cells. In view of these complex activities of Dicer, we tested its antiviral activity in *Myotis myotis* nasal epithelial cells. Surprisingly, we did not see strong evidence of RNAi in these cells, but instead saw a slight proviral effect of Dicer for two alphaviruses, Sindbis and Semliki forest virus. We also observed a striking relocalization of Dicer to cytoplasmic foci upon infection with these viruses, which did not occur in several human cell lines we tested. These foci contained dsRNA and viral plus strand RNA, suggesting that they are sites of viral replication. Importantly, there was no relocalization of Dicer upon infection in *Tadarida brasiliensis* lung epithelial cells that are known to have enhanced RNAi activity, suggesting a link between Dicer localization and antiviral activity. Finally, we found that factors specific to *M. myotis* cells are needed for Dicer relocalization, and that *M. myotis* Dicer has antiviral activity against Sindbis virus when expressed in human cells. Overall, we propose that Dicer can play different roles in distinct bat species and/or cell types, and that its antiviral role appears to be linked to its subcellular localization.

**Data availability statement:** Sequencing data have been deposited on NCBI's Gene Expression Omnibus and are accessible through GEO Series accession number GSE301457.

**Funding:** This work of the Interdisciplinary Thematic Institute IMCbio+, as part of the ITI 2021-2028 program of the University of Strasbourg, CNRS and Inserm, was supported by Agence Nationale de la Recherche through IdEx Unistra (ANR-10-IDEX-0002), SFRI-STRAT'US project (ANR-20-SFRI-0012), and EUR IMCBio (IMCBio ANR-17-EURE-0023) under the framework of the French Investments for the Future Program. It was also supported by an Agence Nationale pour la Recherche PRCI grant to SP (ANR-21-CE35-0018-01). LG's salary was funded by a postdoctoral fellowship from the Fondation pour la Recherche Médicale (funding number SPF202209015746).The funders had no role in study design, data collection and analysis, decision to publish, or preparation of the manuscript.

**Competing interests:** The authors have declared that no competing intersets exist.

## Author summary

Bats are important to study as they are reservoirs for many viruses that can cause severe disease in humans. Notably, bats often show no clinical symptoms when infected with these viruses. How the bat immune system deals with all these viruses and promotes symptom-free infections is still not fully understood, especially in view of the wide diversity of bat species. Here we investigated the antiviral role of the Dicer protein in nasal epithelial cells from the *Myotis myotis* bat species. We find that Dicer behaves differently in these cells compared to human cells or to cells from the *Tadarida brasiliensis* bat species. In *M. myotis* cells, Dicer does not play an antiviral role upon infection with two different alpha-viruses and formed large aggregates at the sites of viral replication. In contrast, Dicer displayed an antiviral activity but did not form cytoplasmic aggregates in *T. brasiliensis* cells. Finally, we show that factors specific to *M. myotis* cells are needed for this phenotype. Uncovering the molecular mechanisms involved in regulating Dicer activity during infection in different bat species will be important to understand how they manage viral infections.

## Introduction

Bats are unique mammals in many aspects, such as their ability to fly and their high longevity relative to their body size [1]. They belong to the order Chiroptera, which can be divided into two suborders that diverged over 50 million years ago [2]. More than 1400 different species of bats have been identified so far, accounting for 20% of all mammals [3]. Importantly, bats are reservoirs for many viruses that can spillover into humans, either directly or through an intermediate host, such as Marburg virus, Hendra and Nipah virus, Middle East respiratory syndrome coronavirus (MERS-CoV), severe acute respiratory syndrome coronavirus (SARS-CoV) and likely SARS-CoV-2 [4–10]. Interestingly, while these viruses can be very pathogenic to humans and other animals, most of the studied bats so far show little to no clinical symptoms, whether they are naturally or experimentally infected [1,11]. An increasing number of studies are highlighting the role of the bat innate immune system in this apparent tolerance to viruses.

The type I and III interferon (IFN) pathways are the main first lines of defense against viruses in mammals. In this pathway, sensing of viral nucleic acids by pattern recognition receptors (PRRs) leads to the expression and secretion of type I and III IFN cytokines. These cytokines are then sensed by cells in an autocrine and para-crine fashion, leading to the expression of interferon stimulated genes (ISGs) that can actively fight the infection [12]. The IFN pathway is present and active in bats, but differences have been reported compared to humans, as well as between different bat species. These include a dampening of the response to DNA sensing, as well as differences in the baseline expression and temporal regulation of some IFNs and ISGs [13–15]. In contrast, invertebrates and plants mostly rely on the RNA interference

(RNAi) pathway to fight viral infections. In antiviral RNAi, viral double-stranded (ds)RNAs are sensed and cleaved into small interfering (si)RNAs by a Dicer or Dicer-like ribonuclease, then loaded into a protein of the Argonaute family to target and degrade complementary viral RNAs [16]. Interestingly, two recent studies have reported enhanced antiviral activity of Dicer through the RNAi pathway in the *Pteropus alecto* kidney (PaKi) cell line infected with Sindbis virus (SINV), an NS1 deletion mutant strain of influenza A virus and a B2 deletion mutant strain of Nodamura virus, and in the *Tadarida brasiliensis* lung (Tb1Lu) cell line infected with SARS-CoV-2 [17,18]. The first study also proposed that viral dsRNA cleavage by Dicer limits the recognition of dsRNA by PRRs and thus the activation of the IFN response in *P. alecto*, thereby suggesting a role for Dicer in bat viral tolerance [17].

Dicer is well known for its essential role in the maturation of the vast majority of micro (mi)RNAs, key regulators of gene expression [19]. In mammals, the same Dicer protein can generate both miRNAs and siRNAs of 22 nucleotides (nt) thanks to its regulatory cofactors transactivation response element RNA-binding protein (TRBP) and Protein ACTivator of the interferon-induced protein kinase (PACT) [20]. However, the antiviral activity of Dicer through the RNAi pathway is controversial and appears to be cell-type and context-dependent in human and mouse cells [21]. Indeed, Dicer isoforms with increased RNAi activity have been identified in human and mouse stem cells [22], and in mouse oocytes [23]. While the former appears to have some antiviral properties against viruses such as Zika and SARS-CoV-2 [22], the mouse oocyte specific isoform does not confer an antiviral phenotype when artificially expressed ubiquitously [24]. One important reason for the reduced antiviral activity of full-length Dicer is the existence of a regulatory crosstalk between the IFN and RNAi pathways [25–27]. Human Dicer is inhibited by the PRR LGP2 (laboratory of genetics and physiology 2/ DexH box polypeptide 58) [28]. Moreover, inactivating the IFN pathway in differentiated mouse cells unmasks long dsRNA processing by Dicer [29]. Conversely, Dicer represses IFN and the ISG protein kinase R (PKR) activity in mouse embryonic stem cells transfected with dsRNA [30]. In addition, we recently showed that human Dicer interacts with PKR *via* its helicase domain and regulates innate immune activation during SINV infection [31,32].

Overall, Dicer could modulate antiviral activity through its role in the RNAi pathway, but also by maturing miRNAs that can affect viral infection, by directly regulating innate immune activation, and potentially through other still unknown mechanisms. How these multiple complex activities of Dicer could contribute to antiviral activity and viral tolerance in different bat species is still poorly understood. This prompted us to investigate the antiviral activity of Dicer in the greater mouse-eared bat (*Myotis myotis*), a European microbat from the *Vespertilionidae* family. Unexpectedly, we only detected a weak RNAi signature in *M. myotis* nasal epithelial (MmNE) cells upon infection with two different viruses. Moreover, we observed a somewhat proviral role of Dicer in these cells upon infection with two different alphaviruses, SINV and Semliki Forest virus (SFV), whereas Dicer played more of an antiviral role as expected in Tb1Lu cells. Interestingly, the proviral activity of Dicer was seemingly correlated with its relocalization to large cytoplasmic foci upon alphavirus infection in MmNE cells, as we did not see relocalization of Dicer in Tb1Lu cells. We find that these foci are likely sites of viral replication and that *M. myotis* cell-specific factors are required for Dicer relocalization and proviral activity. Altogether, we propose that the activity of bat Dicer is context-dependent and could be used by some viruses like SINV in *M. myotis* cells to promote viral replication.

## Results

### *M. myotis* nasal epithelial cells only display a mild RNAi signature

Since other bat cell lines, PaKi and Tb1Lu, were shown to exhibit enhanced antiviral RNAi activity [17,18], we tested if that was also the case in a SV40 transformed *M. myotis* nasal epithelial (MmNE) cell line [33]. We infected these cells with a GFP-expressing SINV (SINV-GFP) [34], which induced antiviral RNAi in PaKi cells [17]. We also tested vesicular stomatitis virus expressing GFP (VSV-GFP) [35], a model rhabdovirus, as many important bat viruses are found within this family [1]. We then sequenced small RNAs isolated from these cells. As a control, we followed the same procedure in human

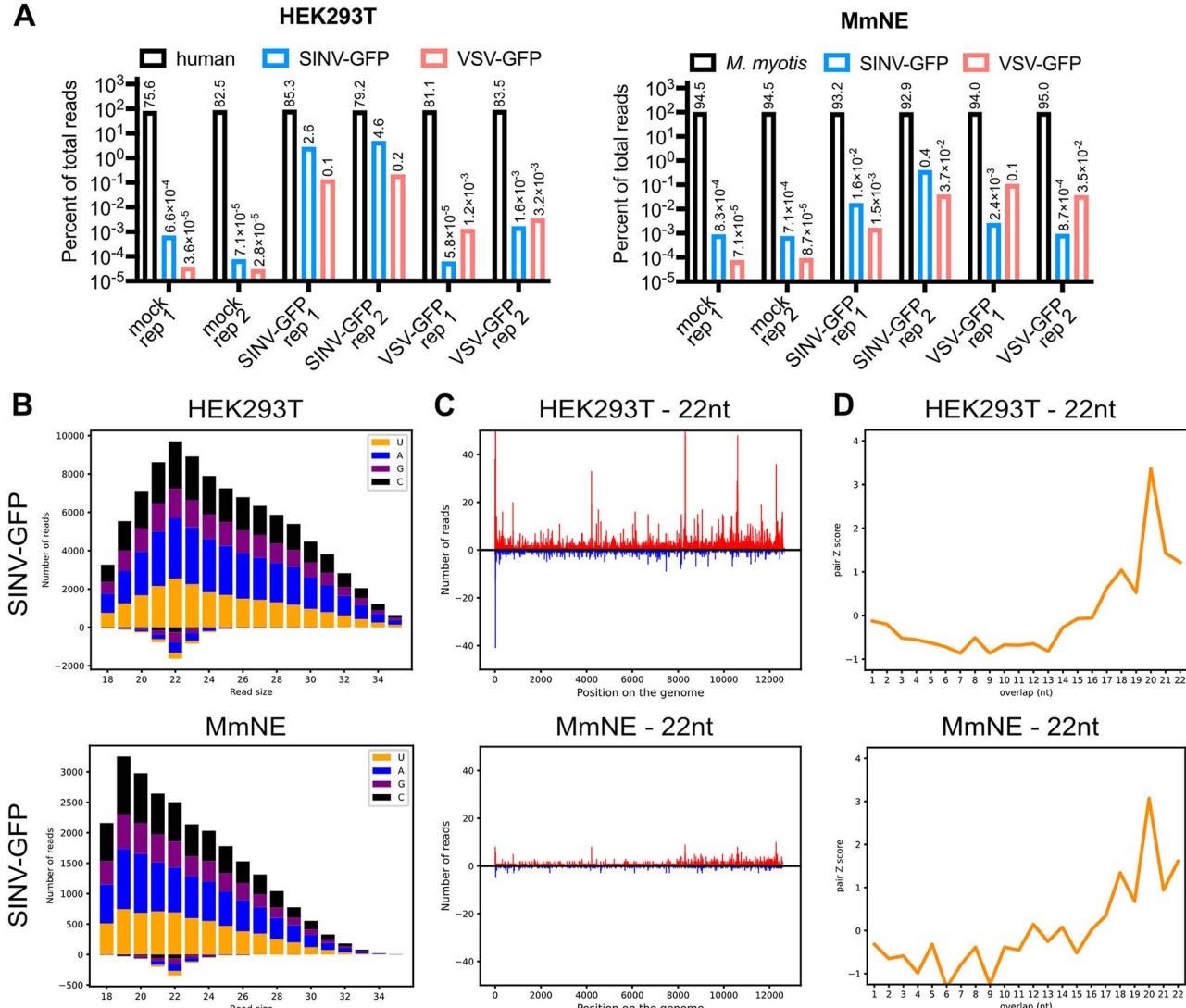

embryonic kidney (HEK)293T cells, which we have previously showed display a very weak RNAi signature against SINV-GFP [36]. Unexpectedly, we did not observe a stronger RNAi signature against SINV-GFP or VSV-GFP in MmNE cells than in HEK293T cells (Fig 1, S1 Fig and S2 Fig). In both HEK293T and MmNE cells, only a low to very low percentage of reads aligned to the SINV-GFP genome, and even lower to the VSV-GFP genome (Fig 1A). Focusing on the SINV-GFP reads, while they showed an enrichment at 22nt on both strands in HEK293T cells, there was an enrichment of smaller

**Fig 1. No strong RNAi signature in *M. myotis* nasal epithelial cells.** HEK293T or MmNE cells were mock infected or infected with SINV-GFP or VSV-GFP, then small RNAs were extracted and sequenced. n = 2 independent experiments, see S1 and S2 Figs for VSV-GFP data and for SINV-GFP data from the other replicate. **(A)** Percent of total sequencing reads that align to the human or *M. myotis* genome (black), the SINV-GFP genome (blue) or the VSV-GFP genome (red) for each sample in HEK293T cells (left) or MmNE cells (right). "rep" indicates the replicate number. **(B)** Number of reads that align to the SINV-GFP genome for each cell type based on the read size. The total number of reads is further broken down into colors based on the identity of the first nucleotide of the read: yellow for U, blue for A, purple for G and black for **C**. **(C)** Location of the 22 nucleotide (nt) reads along the SINV-GFP genome for each sample. The number of reads falling on the same region is represented in red if the reads align to the genome (+ strand) and in blue if they align to the anti-genome (- strand). **(D)** For each sample, 22 nt small RNA pairs that overlap on + and - strands were analyzed and associated Z scores [37] were plotted for the indicated nucleotide overlaps.

size RNAs on the positive strand in the MmNE cells, suggesting the presence of viral RNA degradation products (Fig 1B and S1A Fig). The same enrichment for smaller size RNAs on the positive strand could be seen for VSV-GFP reads in MmNE cells, while almost no reads aligned to the VSV-GFP genome in HEK293T cells (S2A Fig). The few viral reads that were 22nt long did display typical siRNA characteristics in both cell lines, as they aligned to both strands and throughout the SINV genome (Fig 1C, S1B Fig and S2B Fig). For each sequence of 22nt aligning to the viral plus strand, we also calculated the probability of pairing with another 22nt sequence aligning to the viral minus strand, and plotted this probability as a Z score based on the number of overlapping nucleotides (Fig 1D and S1C Fig) [37]. In both cell lines infected with SINV-GFP, the Z score peaked for an overlap of 20nt, indicative of an enrichment for paired SINV-GFP reads that overlap with a 2nt overhang, which is a signature of Dicer cleavage as part of the RNAi pathway (Fig 1D and S1C Fig). For VSV-GFP reads, the number of 22nt pairs was so low that the Z scores were not meaningful. Altogether, our results indicate that there is some evidence of antiviral RNAi activity against SINV-GFP in MmNE cells, but the very low total number of viral reads of 22nt overall and compared to HEK293T cells argues against an enhanced antiviral RNAi activity of Dicer in MmNE cells.

## Dicer does not protect against two alphaviruses in *M. myotis* nasal epithelial cells

To test whether the weak RNAi signature we observed in MmNE cells could nevertheless result in antiviral activity, we studied the effect of knocking down Dicer on viral replication. We first tested two individual siRNAs against *M. myotis* Dicer (hereafter referred to as mmDicer). They were both able to decrease mmDicer protein levels, although the knock down efficiency was limited (S3A Fig). Consistent with the weak RNAi signature, we did not see any antiviral activity of mmDicer, but instead a somewhat proviral effect against SINV-GFP infection. Indeed, knocking down Dicer in MmNE cells led to a slight decrease in SINV-GFP RNA, GFP protein and SINV-GFP titers, although not always significant (S3A - S3C Fig). Since the lack of effect could be due to the poor knock down efficiencies of our individual Dicer siRNAs, we turned to a custom designed pool of 30 siRNAs (siPool, siTOOLs Biotech) to knock down Dicer more efficiently in MmNE cells. This technique allows to keep siRNAs at low concentrations (3 nM) to limit off-target effects, while providing a very potent Dicer knock down (Fig 2A). Again, we did not see any antiviral activity of mmDicer in MmNE cells against SINV-GFP when quantifying viral titers over time and with two different starting multiplicity of infection (MOIs) (S3D Fig). Instead, we saw a decrease in GFP+ cells over time upon Dicer knock-down (Fig 2B). At 24 hours post infection, there was again a slight decrease in GFP protein levels and in SINV-GFP RNA levels upon Dicer knock down (Fig 2C - 2D), which had a measurable but not significant impact on SINV-GFP titers (Fig 2E). Overall, our results suggest that Dicer does not protect against SINV-GFP infection in MmNE cells.

To check whether the somewhat proviral effect of Dicer could be recapitulated with other viruses, we tested the effect of siPOOL transfection on another alphavirus, SFV, as well as on VSV-GFP infection. Similar to SINV-GFP infection, knocking down Dicer significantly decreased SFV RNA levels at 24 hours post infection, which resulted in a small but not significant impact on SFV titers (Fig 2F - 2G). In contrast, Dicer knock down had a proviral effect against VSV-GFP infection at the RNA and protein levels, but no difference in titers was observed (Fig 2H - 2J). Altogether, our results suggest that unlike described for PaKi and Tb1Lu cells, Dicer does not display enhanced antiviral RNAi activity in MmNE cells, but instead plays a moderate proviral role for SINV-GFP and SFV but not VSV-GFP infection.

## Dicer relocalizes to cytoplasmic foci upon SINV infection in *M. myotis* nasal epithelial cells

To try to understand how Dicer could play a proviral role in MmNE cells, we looked at the subcellular localization of Dicer upon SINV infection in MmNE cells. We made sure that our Dicer antibody could specifically recognize both hDicer and mmDicer by immunofluorescence microscopy (S4 Fig). First, knocking down Dicer with siPool in MmNE cells significantly decreased the overall level of Dicer staining (S4A - S4B Fig). Second, we tested HEK293T cells knock-out for Dicer (NoDice) cells [38]) that stably express Flag-HA-tagged human Dicer (hDicer) or an empty vector (EV). We also cloned

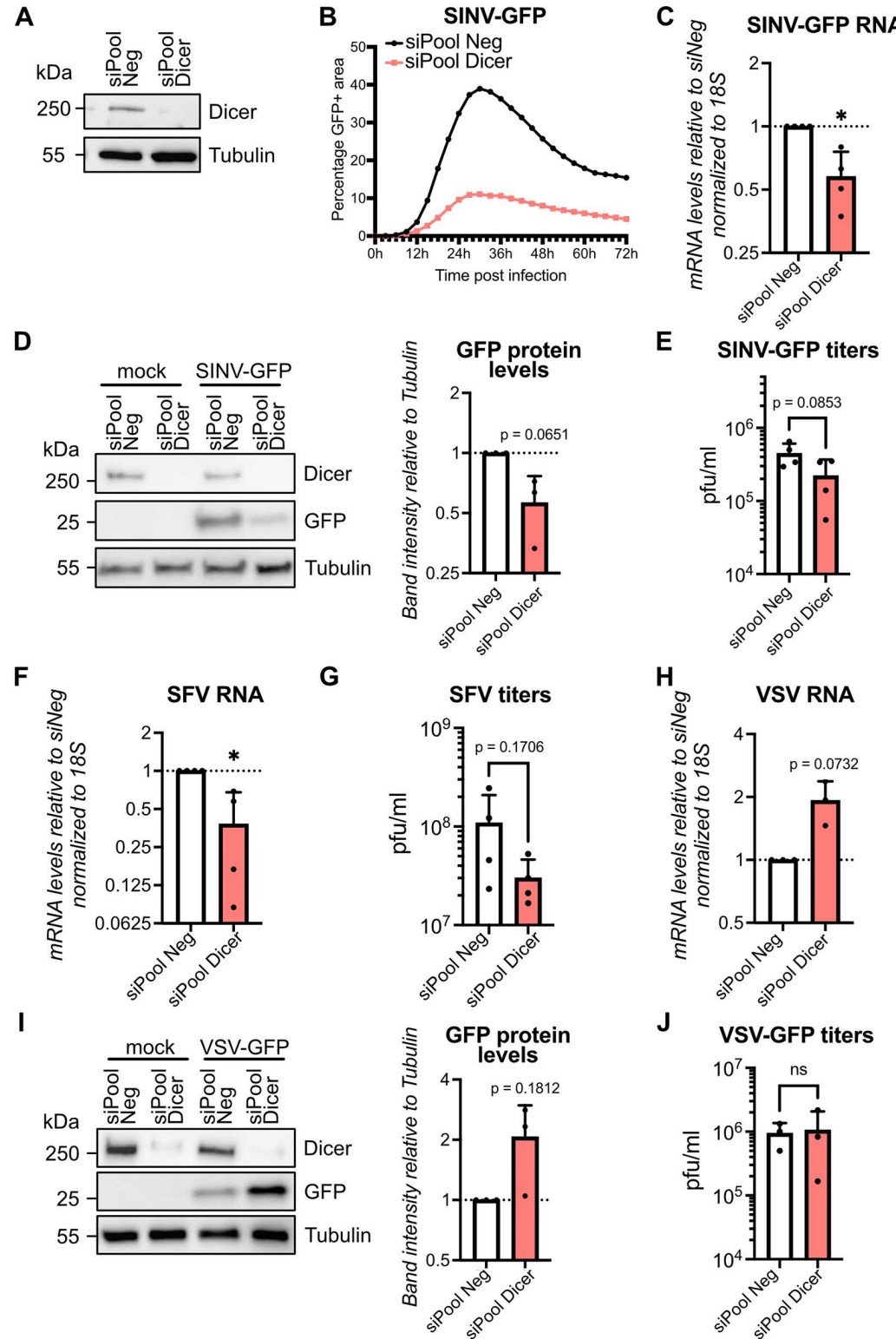

**Fig 2. Dicer does not protect against SINV and SFV in *M. myotis* nasal epithelial cells. (A)** Expression of Dicer was analyzed by western blot in MmNE cells after two rounds of siPool siRNAs targeting Dicer (siPool Dicer) or a non-targeting control (siPool Neg). α-Tubulin was used as loading control. Images are representative of 2 independent experiments. **(B-J)** MmNE cells treated with siPool siRNAs targeting Dicer (siPool Dicer) or a non-targeting control (siPool Neg) were infected with SINV-GFP for 24h at MOI 0.2 **(B-E)**, SFV for 24h at MOI 20 (F-G) or VSV-GFP for 8h at MOI 1

(H-J). (B) The number of GFP+ cells as a proxy for infection were monitored over time by live microscopy imaging. The percentages of GFP+ area were plotted for each condition as the average of two wells, six photos per well, for one experiment. (C, F, H) RNAs were purified and levels of SINV-GFP RNA (C), SFV RNA (F) or VSV-GFP RNA (H) were quantified by qRT-PCR. Means+standard deviations were plotted relative to siPool Neg infected samples for 3 (VSV) or 4 (SINV and SFV) independent experiments. (D, I) Protein lysates were collected and analyzed by western blot using antibodies for Dicer and GFP. α-Tubulin was used as loading control. Images are representative of 3 independent experiments. GFP band intensities relative to Tubulin band intensities were quantified with ImageJ for each experiment, and means+standard deviations were plotted relative to siPool Neg infected samples. (E, G, J) Supernatants were collected and viral titers were quantified by plaque assay. Means+standard deviations were plotted for 3 (VSV) or 4 (SINV and SFV) independent experiments. For all RNA and protein graphs, p values were calculated using one sample t tests compared to 1. For all viral titer graphs, p values were calculated using unpaired t tests. * p<0.05; ns: not significant.

mmDicer from MmNE cells and expressed it with Flag and HA tags in NoDice cells. The level of Dicer staining by immunofluorescence microscopy was higher in NoDice cells expressing hDicer or mmDicer compared to an empty vector control (S4C - S4D Fig). Moreover, Flag staining followed the same pattern as Dicer staining in NoDice cells expressing hDicer or mmDicer with a high Pearson correlation coefficient (S4C Fig). Taken together, these results confirmed that our Dicer antibody could specifically detect mmDicer and can be used for immunofluorescence microscopy.

We thus used this antibody to study the localization of Dicer in MmNE cells. In mock conditions, mmDicer exhibited a diffuse cytoplasmic localization in MmNE cells (Fig 3). Interestingly, upon WT SINV infection we saw a striking relocalization of Dicer into discrete cytoplasmic foci. This relocalization could already be noticed at 8h (Fig 3A) and was still apparent at 24h (Fig 3B). There was no significant difference in both the number and size distribution of Dicer foci between 8h and 24h, although there was a small trend towards a decrease in the number of Dicer foci per cell (Fig 3C). Importantly, we did not observe such a shift in Dicer localization upon SINV infection in HEK293T cells (Fig 3A - 3B), as well as for two other human cell lines we tested, human lung epithelial carcinoma A549 cells and human hepatocyte-derived carcinoma Huh7 cells (S5A Fig). Overall, our results suggest a singular activity of Dicer during SINV infection in MmNE cells that leads to its relocalization in large cytoplasmic foci, but that does not appear to occur in human cells.

## Dicer likely relocalizes to sites of alphavirus replication in *M. myotis* nasal epithelial cells

SINV infected cells are known to accumulate large quantities of dsRNA molecules in the cytoplasm as a result of replication [39]. Since dsRNA is the primary substrate of Dicer, we wondered if Dicer clustered around dsRNA in SINV infected cells. We infected MmNE cells with WT SINV and measured the co-localization of Dicer and dsRNA by confocal microscopy, using J2 antibodies that recognize dsRNAs longer than 40 bp [40]. We saw that many of the Dicer foci co-localized with dsRNA in SINV infected cells (Fig 4A), with a high Pearson correlation coefficient across multiple experiments (Fig 4B), suggesting that Dicer forms foci in close proximity to dsRNA in MmNE cells. In human cells, we could also see dsRNA accumulating in SINV infected HEK293T, A549 and Huh7 cells but Dicer staining remained diffuse and did not co-localize with dsRNA (Fig 4A - 4B and S5A Fig). Importantly, infection with SFV also led to Dicer foci that co-localized with dsRNA in MmNE cells (Fig 4A - 4B). In contrast, it has been shown that dsRNA does not accumulate to detectable levels during VSV infection [41]. Accordingly, we did not detect dsRNA and there was no relocalization of Dicer upon VSV infection in MmNE or HEK293T cells (S5B Fig). These results suggest that Dicer does not form cytoplasmic foci for every viral infection but its relocalization may be specific to alphaviruses or linked to dsRNA accessibility. Accumulation of dsRNA is a hallmark of viral replication, we thus wondered if Dicer was present at sites of SINV replication. Since J2 antibodies cannot distinguish between dsRNA of viral origin or originating from the host, we carried out fluorescence *in situ* hybridization (FISH) to specifically stain for SINV plus strand RNA. As expected, we could visualize the accumulation of SINV RNA in the form of discrete foci, most likely corresponding to sites of viral replication, in both HEK293T cells and MmNE cells (Fig 4C and S5C Fig). Importantly, Dicer stayed diffuse upon infection in HEK293T cells whereas the Dicer foci matched the SINV plus strand RNA foci in MmNE cells (Fig 4C and S5C Fig). Altogether, our results suggest that Dicer is recruited to sites of SINV replication in MmNE cells.

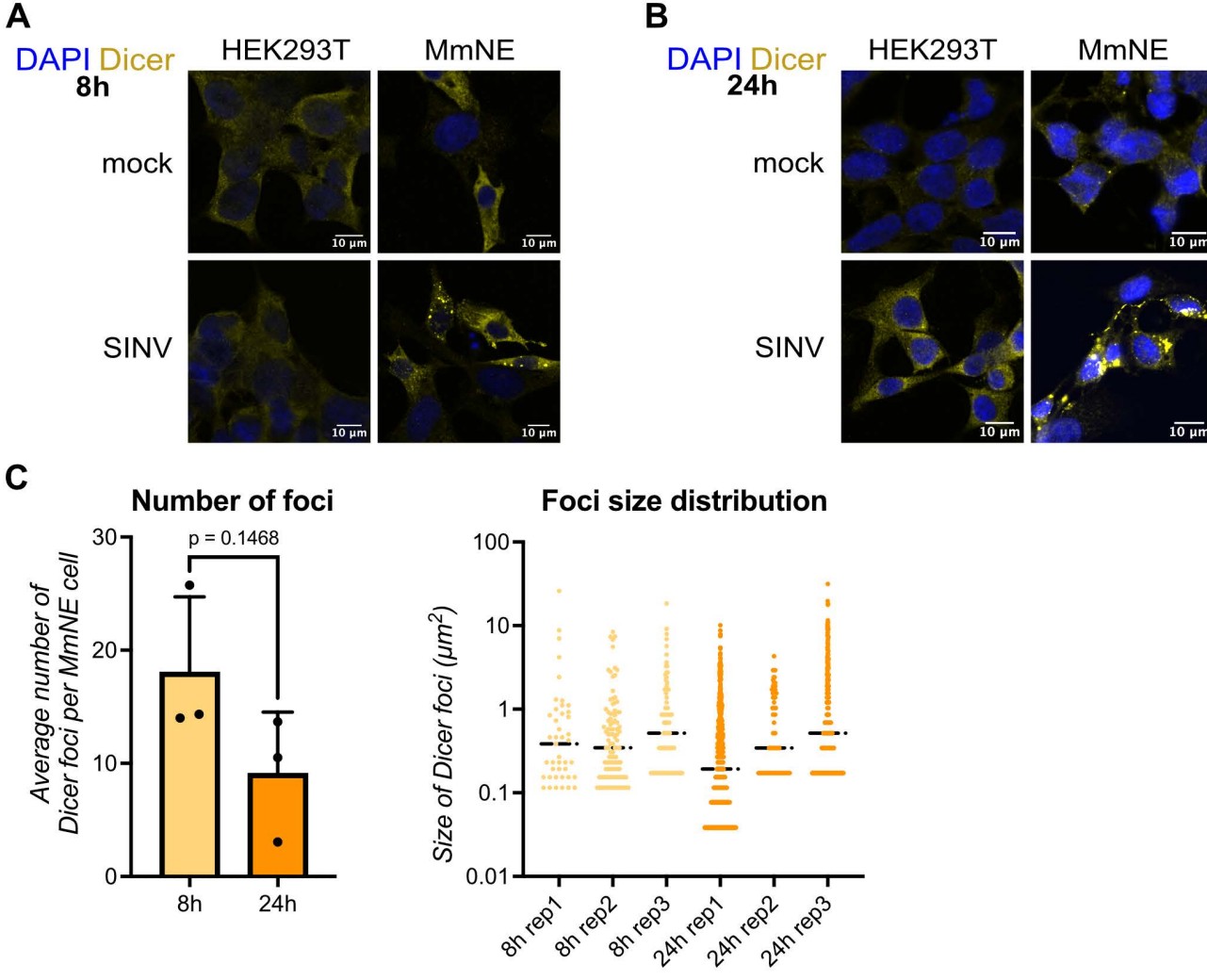

**Fig 3. Dicer relocalizes to distinct cytoplasmic foci upon infection in *M. myotis* nasal epithelial cells.** HEK293T were mock infected or infected with WT SINV for 8h at MOI 0.2 (A) or 24h at MOI 0.02 **(B)**. MmNE cells were mock infected or infected with WT SINV for 8h at MOI 5 (A) or 24h at MOI 2 **(B)**. Localization of Dicer (yellow) was imaged by immunofluorescence and confocal microscopy. DAPI staining (blue) indicates cell nuclei. Images are representative of at least 3 independent experiments. Scale bars 10μm. **(C)** For each condition, the number and size of Dicer foci in SINV infected MmNE cells were plotted for 3 independent experiments with each 1 or 2 fields containing approximately the same number of cells. The black horizontal bars represent the median. The p value was calculated using an unpaired t test.

## Dicer does not relocalize to sites of SINV replication in *T. brasiliensis* lung cells

So far, we have observed a lack of Dicer-mediated antiviral RNAi activity against SINV infection in MmNE cells, as well as the relocalization of Dicer to sites of SINV replication in these cells. However, we do not know whether the two phenomena are linked. To try to answer this question, we decided to study the localization of Dicer in a bat cell line that was shown to have enhanced antiviral RNAi activity [17,18]. We focused on the *Tadarida brasiliensis* (tb) lung epithelial Tb1Lu cell line as it is commercially available. While our Dicer siPool siRNAs were specific to MmNE cells, the individual Dicer siRNAs we designed can also recognize tbDicer mRNA. We therefore tested those in Tb1Lu cells and observed that they were able to decrease Dicer protein levels (Fig 5A). Unlike in MmNE cells, Dicer did not have a proviral effect but a measurable antiviral effect against SINV-GFP in Tb1Lu cells, consistent with its previously reported antiviral activity against

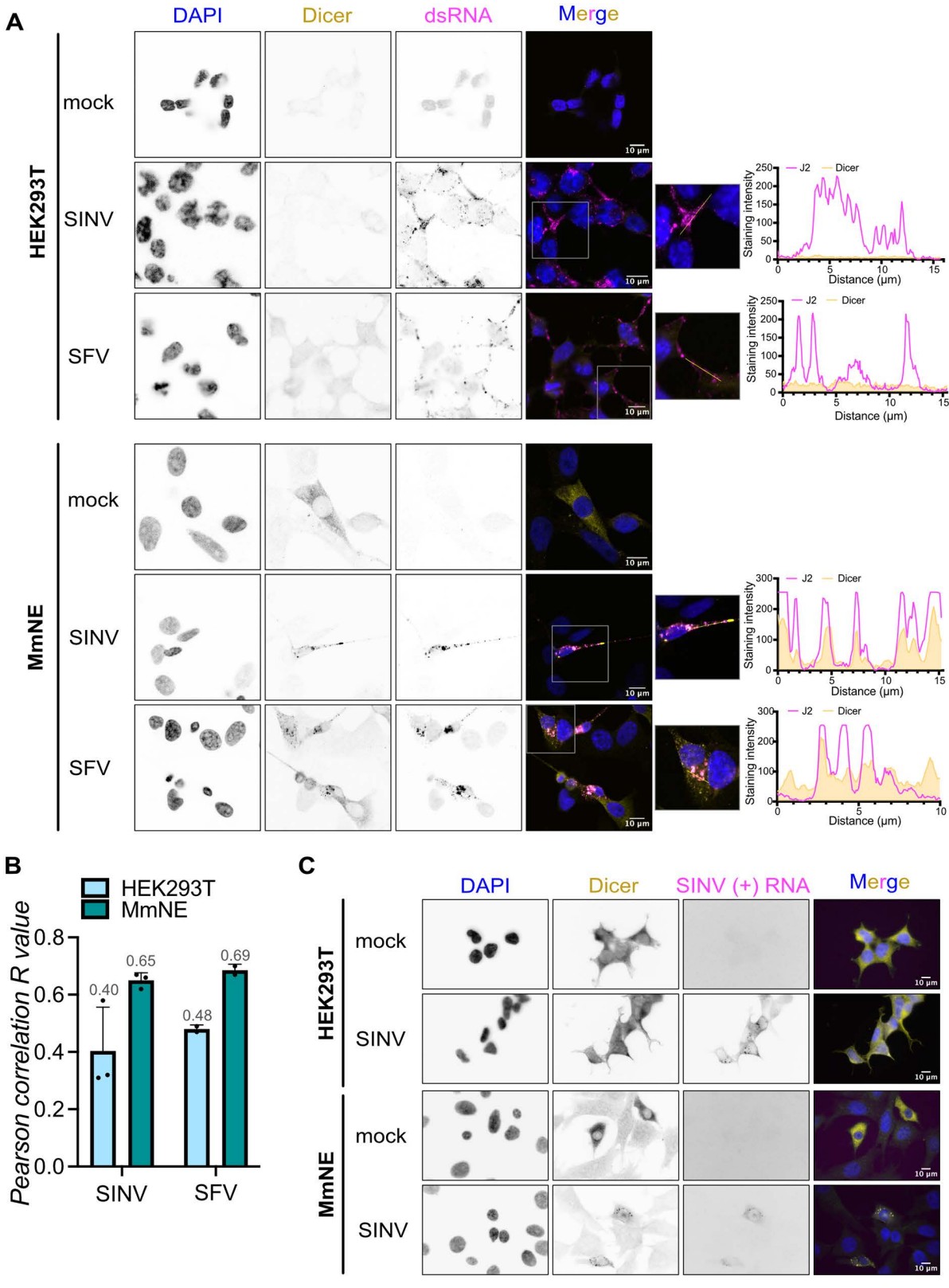

**Fig 4. Dicer likely relocalizes to sites of SINV replication in *M. myotis* nasal epithelial cells. (A-B)** HEK293T were mock infected or infected with WT SINV or SFV for 24h at MOI 0.02. MmNE cells were mock infected or infected with WT SINV or SFV for 24h at MOI 2 or 5 respectively. Localization of Dicer (yellow) and dsRNA via J2 antibodies (magenta) was imaged by immunofluorescence and confocal microscopy. DAPI staining (blue) indicates cell nuclei. Images are representative of 3 (SINV) or 2 (SFV) independent experiments. Scale bars 10µm. Co-localization between Dicer and dsRNA signals was quantified and Pearson correlation R coefficients were calculated and plotted for 3 (SINV) or 2 (SFV) independent experiments – mean (numerical value indicated in gray font above each bar) + standard deviation **(B)**. A zoomed-in image of the boxed area in some merge images was added. The graphs show the signals for each channel along the line drawn in the zoomed in images. **(C)** HEK293T or MmNE cells were mock infected or infected with WT SINV for 24h at MOI 0.02 or 4, respectively. Localization of Dicer (yellow) by immunofluorescence and of SINV plus strand (+) RNA (magenta) by FISH was imaged with an epifluorescence microscope. See S5C Fig for confocal microscopy images of the same samples. DAPI staining (blue) indicates cell nuclei. Images are representative of 2 independent experiments. Scale bars 10µm.

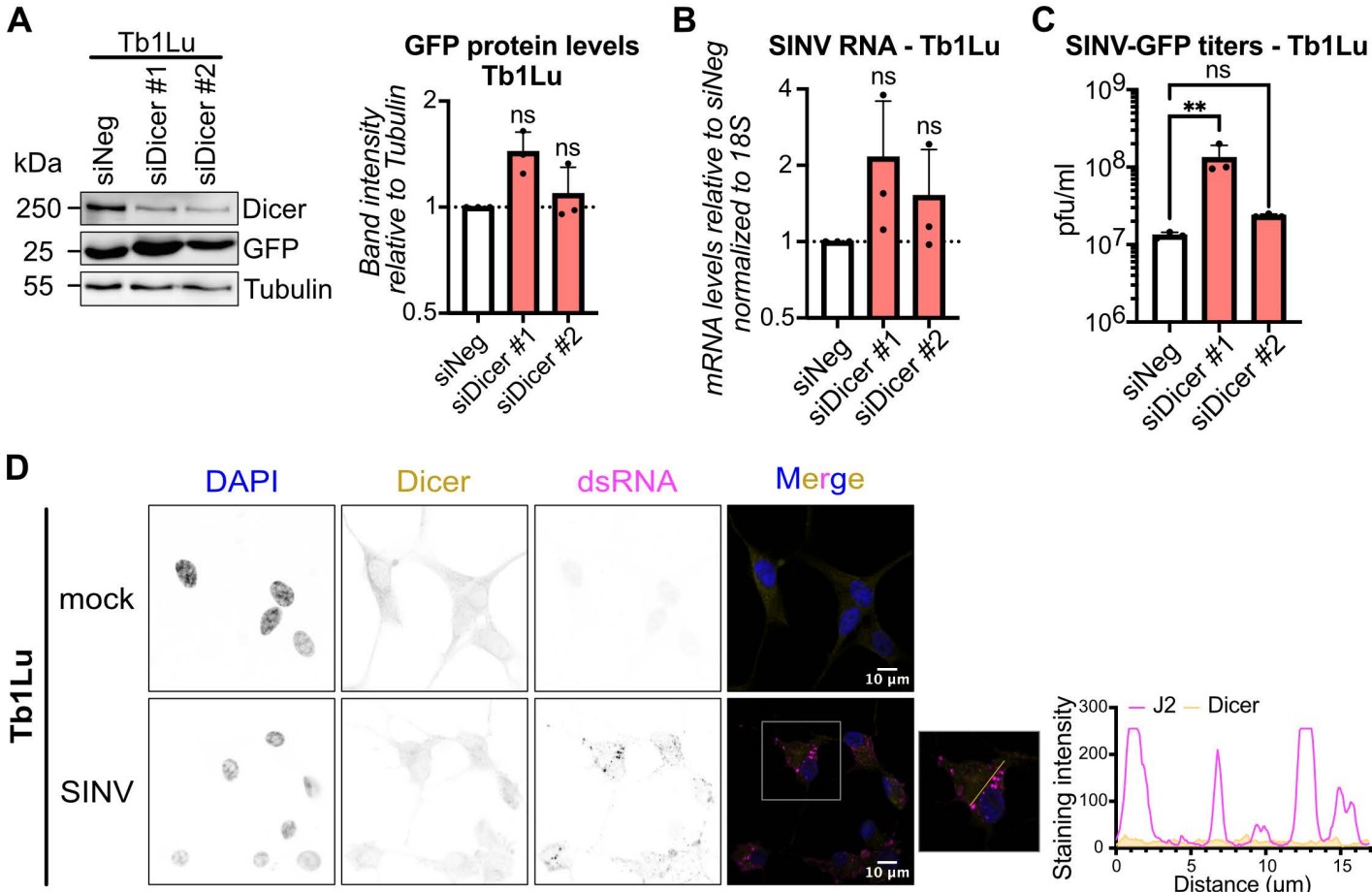

**Fig 5. Dicer is antiviral and does not relocalize to sites of SINV replication in *T. brasiliensis* lung cells. (A-C)** Tb1Lu cells treated with two different siRNAs targeting Dicer (siDicer) or a non-targeting control (siNeg) were infected with SINV-GFP for 24h at MOI 0.02. **(A)** Protein lysates were collected and analyzed by western blot using antibodies for Dicer and GFP. α-Tubulin was used as loading control. Images are representative of 3 independent experiments. GFP band intensities relative to Tubulin band intensities were quantified with ImageJ for each experiment, and means + standard deviations were plotted relative to siNeg infected samples. **(B)** RNAs were purified and levels of SINV-GFP RNA were quantified by qRT-PCR. Means + standard deviations were plotted relative to siNeg infected samples for 3 independent experiments. **(C)** Supernatants were collected and viral titers were quantified by plaque assay. Means + standard deviations were plotted for 3 independent experiments. For the RNA and protein graphs, p values were calculated using one sample t tests compared to 1. For the viral titer graph, p values were calculated using an ordinary one-way ANOVA with Dunnett's multiple comparison test with a single pooled variance compared to siNeg. ** p < 0.01; ns: not significant. **(D)** Tb1Lu cells were mock infected or infected with WT SINV for 24h at MOI 0.02. Localization of Dicer (yellow) and dsRNA via J2 antibodies (magenta) was imaged by immunofluorescence and confocal microscopy. DAPI staining (blue) indicates cell nuclei. Images are representative of 2 independent experiments. Scale bars 10µm. A zoomed-in image of the boxed area in the SINV merge image was added. The graph shows the signals for each channel along the line drawn in the zoomed in image.

SARS-CoV-2 in these cells [18]. Indeed, knocking down Dicer in Tb1Lu cells led to a slight increase in SINV-GFP RNA, GFP protein and SINV-GFP titers at 24 hours post infection, with a stronger effect for one of the two siRNAs (Fig 5A - 5C). Importantly, while dsRNA accumulated as expected in SINV infected Tb1Lu cells, there was no apparent Dicer relocalization into cytoplasmic foci in these cells (Fig 5D), resulting in a low Pearson correlation coefficient between Dicer and dsRNA (0.43 and 0.47 for the two independent experiments). These results indicate that Dicer does not relocalize to sites of viral replication upon SINV infection in Tb1Lu cells. These results also suggest a possible inverse correlation between Dicer relocalization and its antiviral activity.

### *M. myotis* Dicer expressed alone in human NoDice cells does not relocalize to sites of SINV replication and protects from SINV infection

Finally, we wanted to know whether there was something inherently special about mmDicer that triggered its relocalization upon SINV infection. To this end, we characterized the localization and antiviral activity of mmDicer outside of its native context, *i.e.,* when expressed in human NoDice cells. As controls, we also expressed Flag-HA-tagged human Dicer (hDicer) or an empty vector (EV). Efficient expression of the different Dicer constructs was verified by western blot (Fig 6A). Expression of either hDicer or mmDicer rescued maturation of the human miRNA miR-16 (Fig 6B). These results suggested that mmDicer is indeed expressed and active in NoDice cells. We then determined the localization of Dicer upon SINV infection in NoDice cells. As expected, while J2 staining showed discrete foci characteristic of SINV infection, hDicer stayed diffuse throughout the cytoplasm (Fig 6C). Interestingly, there was also no relocalization of mmDicer upon SINV infection (Fig 6C). Importantly, relocalization of Dicer did not correlate with its abundance, as mmDicer was overexpressed in NoDice:mmDicer cells compared to MmNE cells (S6A Fig). These results suggested that mmDicer alone is not sufficient to form large foci, and that factors specific to MmNE cells are required for relocalization. Of note, immunoprecipitating mmDicer in these cells showed that mmDicer still interacted with known hDicer interactors such as PACT and TRBP in mock conditions, as well as PKR and ADAR1, which we previously showed interact with hDicer during SINV infection [31] (S6B Fig).

We then tested the antiviral activity of mmDicer against SINV-GFP and VSV-GFP. Consistent with previous results, expressing mmDicer did not protect against VSV-GFP infection over time (S6C Fig). Unexpectedly, expressing mmDicer decreased SINV-GFP titers over time for two different starting MOIs compared to the empty vector control and hDicer (S6D Fig), suggesting that mmDicer can protect again SINV-GFP infection outside of MmNE cells. To confirm these results, we picked one MOI (0.02) and one time point (24h) and quantified SINV-GFP titers, protein and RNA levels. As controls, we cloned tbDicer from Tb1Lu cells and expressed it with Flag and HA tags in NoDice cells. We also expressed Flag-HA-tagged catalytically inactive mutant versions of mmDicer (mmDicer CM, amino acid changes E1562A and E1819A) and tbDicer (tbDicer CM, amino acid changes E1559A and E1808A). These new constructs were efficiently expressed (Fig 6A), and expressing tbDicer but not tbDicer CM or mmDicer CM rescued maturation of miR-16 in NoDice cells (Fig 6B). VSV-GFP RNA, protein and titer levels were unchanged upon expression of mmDicer or tbDicer (S6E - S6G Fig). In contrast, expressing mmDicer or tbDicer decreased SINV-GFP RNA, protein and titer levels to varying degrees (Fig 6D - 6F). Interestingly, mmDicer CM also exhibited some antiviral activity against SINV-GFP compared to hDicer, which needs to be further explored (Fig 6D - 6F). These results confirm that mmDicer is antiviral against SINV-GFP when expressed in human NoDice cells, and again suggests an inverse correlation between Dicer relocalization and its antiviral activity.

### Discussion

Recent studies have suggested a role for Dicer in bat antiviral immunity and tolerance to viruses through RNAi in two different bat species, *P. alecto* and *T. brasiliensis* [17,18]. In this study, we set out to characterize the antiviral activity of Dicer in a *M. myotis* nasal epithelial cell line. Interestingly, MmNE cells did not mount a strong RNAi antiviral response against

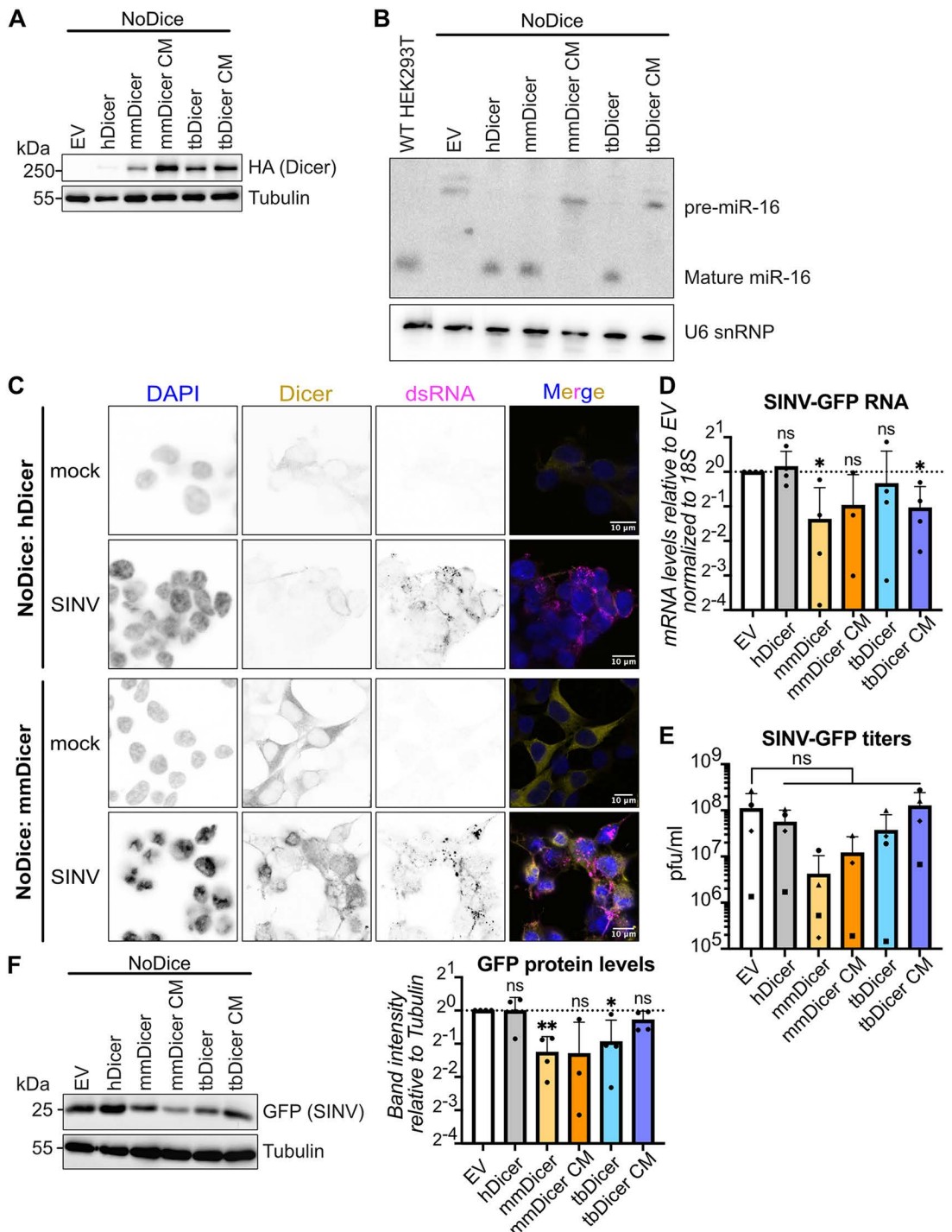

**Fig 6. *M. myotis* Dicer expressed alone in human NoDice cells does not relocalize to sites of SINV replication and protects from SINV infection.** (A) Expression of HA-tagged Dicer was analyzed by western blot for HEK293T NoDice cells expressing Flag-HA tagged hDicer, mmDicer, mmDicer CM, tbDicer, tbDicer CM or an empty vector control. α-Tubulin was used as loading control. Images are representative of 3 independent experiments. (B) Expression of miR-16 was analyzed by northern blot for HEK293T NoDice cells expressing Flag-HA tagged hDicer, mmDicer, mmDicer CM, tbDicer, tbDicer CM or an empty vector control. A probe for U6 snRNP was used as loading control. (C) HEK293T NoDice cells expressing Flag-HA tagged hDicer or mmDicer were mock infected or infected with SINV-GFP for 24h at MOI 0.02. Localization of Dicer (yellow) and dsRNA via J2 antibodies (magenta) was imaged by immunofluorescence and confocal microscopy. DAPI staining (blue) indicates cell nuclei. Images are representative of 2

independent experiments. Scale bars 10μm. (D-F) HEK293T NoDice cells expressing Flag-HA tagged hDicer, mmDicer, mmDicer CM, tbDicer, tbDicer CM or an empty vector control were infected with SINV-GFP at MOI 0.02 for 24h. (D) RNAs were purified and levels of SINV-GFP RNA were quantified by qRT-PCR. Means + standard deviation normalized to 18S were plotted relative to empty vector control for 3 (mmDicer CM) or 4 independent experiments. (E) Supernatants were collected and viral titers were quantified by plaque assay. Means + standard deviations were plotted for 3 (mmDicer CM) or 4 independent experiments. Each symbol represents a separate experiment. (F) Protein lysates were collected and analyzed by western blot using antibodies for GFP. α-Tubulin was used as loading control. Images are representative of 3 (mmDicer CM) or 4 independent experiments. GFP band intensities relative to Tubulin band intensities were quantified with ImageJ and the mean + standard deviation was plotted relative to empty vector control for each experiment. For the RNA and protein graphs, p values were calculated using one sample t tests compared to 1. For the viral titer graph, p values were calculated using an ordinary one-way ANOVA with Dunnett's multiple comparison test with a single pooled variance compared to EV. * $p < 0.05$; ** $p < 0.01$; ns: not significant.

SINV-GFP and VSV-GFP infection. Instead, knocking down Dicer in MmNE cells led to a slight decrease in SINV-GFP and SFV replication, while results were more mitigated with VSV. In addition, we observed a striking relocalization of Dicer in large cytoplasmic foci upon SINV and SFV infection in MmNE cells. These Dicer foci are in close proximity with dsRNA and SINV plus strand RNA, suggesting that Dicer is recruited to sites of SINV replication. In contrast, no Dicer foci were observed upon infection in several human cell lines. Importantly, the *T. brasiliensis* Tb1Lu cell line that was found to have enhanced RNAi behaved differently. Knocking down Dicer in these cells led to a slight increase in SINV-GFP replication, and no Dicer relocalization was observed. Finally, mmDicer does not form these foci when expressed in human NoDice cells, suggesting that its relocalization requires specific factors that are present in MmNE cells. Moreover, mmDicer exhibited antiviral activity against SINV-GFP when expressed in NoDice cells, suggesting an inverse correlation between Dicer relocalization and its antiviral activity.

Our results that mmDicer does not have increased antiviral RNAi activity in MmNE cells is unexpected in view of the recently published results on *P. alecto* and *T. brasiliensis* Dicer, especially as SINV was also used in the *P. alecto* study [17,18]. This discrepancy could come from differences in bat species, as *P. alecto* and *M. myotis* are thought to have diverged more than 60 million years ago, and *T. brasiliensis* and *M. myotis* at least 35 million years ago [42–44]. Yet, Dicer is well conserved across bats and mammals, with 96% amino acid conservation between mmDicer and tbDicer and 95% between mmDicer and *P. alecto* Dicer (S7 Fig). Since in human and mouse cells the antiviral RNAi activity of Dicer can be cell-type dependent, it is also possible that the activity of bat Dicer is influenced by the cell type used in each study, especially if different bat species and cell types have differences in IFN activity [15,45,46]. In general, it will be important to study the activity of Dicer in more bat species to resolve this discrepancy.

One caveat of our study is that we do not have a direct comparison of the viral siRNA abundance in SINV-infected MmNE and Tb1Lu cells. However, it is important to point out that increased siRNA abundance does not necessarily mean increased antiviral RNAi activity. Indeed, we saw more viral small RNAs with an siRNA signature against SINV in HEK293T cells (Fig 1 and S1 Fig), even though Dicer is not antiviral against SINV in these cells [36], something we confirmed by re-expressing human Dicer in NoDice cells (Fig 6D - 6F). A better approach would be to use a reporter of RNAi activity [29,47], or to quantify the viral siRNAs that are actually loaded into RNA-induced silencing complexes since loading can be inefficient, leading to little antiviral activity as shown for influenza A virus [48]. Nonetheless, knocking down Dicer led to opposite results in MmNE and Tb1Lu cells. While this argues against enhanced antiviral RNAi activity for MmNE cells, the apparent antiviral activity of Tb1Lu cells against SINV could be through RNAi and/or through another activity of Dicer. In mammalian cells, the same Dicer protein could play a role in antiviral immunity through its RNAi activity, but also through direct regulation of the innate immune response and through the maturation of miRNAs that regulate the innate immune response [26,27]. These multiple interconnected activities of Dicer make the study of Dicer antiviral activity difficult. We employed a transient siRNA knockdown strategy instead of knocking out Dicer to limit the impact of removing Dicer on the expression of miRNAs, which have a notoriously long half-life [49], but there could still be effects of Dicer on short-lived miRNAs influencing its propensity to be anti- or proviral in specific cellular contexts. It will be important

to tease apart the different activities of Dicer, especially as mmDicer CM seems to maintain some antiviral activity against SINV in NoDice cells (Fig 6D - 6F).

The case of VSV is interesting, as knocking down Dicer in *M. myotis* cells led to an increase in VSV-GFP RNA and proteins levels (Fig 2H - 2I). Yet, there was almost no viral siRNAs (S2 Fig). The mechanism behind this apparent antiviral activity of Dicer is thus unclear, although overall there was no impact on titer levels. Nonetheless, it serves as a good control of a virus that does not trigger relocalization of Dicer in MmNE cells, suggesting that this relocalization is context-specific and not an artefact of infection. Since dsRNA is not detectable upon VSV infection, it is tempting to hypothesize that the relocalization of Dicer is linked to dsRNA accessibility. It will be interesting to test this hypothesis.

Dicer relocalization to discrete cytoplasmic foci upon infection in MmNE cells is reminiscent of *Drosophila* D2 bodies [50]. These D2 bodies are cytoplasmic foci containing Dicer-2 and its co-factor R2D2 that are important for the processing of endogenous siRNAs in *Drosophila* cells. Yet, our small RNAseq results did not identify such increased activity of Dicer. Instead, our results in Tb1Lu and NoDice cells argue for an inverse correlation between Dicer relocalization and its antiviral activity, although a mechanism is needed to confirm that the two phenomena are directly linked. We can nonetheless propose several hypotheses as to how the relocalization of Dicer could play a proviral role. Potential consequences of this relocalization could be the sequestration of Dicer by SINV, for example to prevent Dicer activity as part of the antiviral RNAi pathway. Alternatively, Dicer could be recruited to sites of SINV replication to hide dsRNA from PRRs or to directly tone down the local innate immune response. Dicer could also have a structural role to help viral factory formation and/or alphavirus replication. Possible explanations for this recruitment, which remain to be tested, could be the existence of *M. myotis* specific co-factors, or differences in Dicer co-factors between species. mmDicer could also have a higher affinity for dsRNA in MmNE cells. More work will be needed to identify the purpose of Dicer relocalization and the link between this relocalization and Dicer proviral activity.

It is intriguing that this relocalization only appears in MmNE cells, and it will be important to test cell lines from other more closely related species. The fact that mmDicer alone is not sufficient for relocalization is consistent with mmDicer and hDicer being very similar, with 95% conservation between the two proteins at the amino acid level (S7 Fig). On top of finding the mechanism of Dicer relocalization, it will also be important to determine whether Dicer forms cytoplasmic foci during infection with other viruses and in other stress conditions. While many different alphaviruses have been detected in several species of bats in the wild, including SINV and SFV, it is still not clear whether bats constitute a reservoir for alphaviruses as this has not been well studied [51,52]. It will thus be important to also test other viruses that are more biologically relevant to *M. myotis*.

## Materials and methods

### Cell culture and viral infections

Vero E6 and A549 cell lines were obtained through the American Type Culture Collection (ATCC, cat# CRL-1586 and CCL-185, respectively). SV40 transformed *Myotis myotis* nasal epithelial cells [33] were a kind gift from Bernd Köllner (FLI, Greifswald, Germany) through Lucie Etienne (CIRI, Lyon, France). HEK293T and HEK293T/NoDice (2.20) cells [38] were a kind gift from Bryan Cullen (Duke University, Durham NC, USA). *Tadarida brasiliensis* lung epithelial Tb1Lu cells (also available from ATCC, cat# CCL-88) were a kind gift from Nolwenn Jouvenet (Institut Pasteur, Paris, France). Huh7.5.1 (Huh7) cell lines were a kind gift from Thomas Baumert (Institute for Translational Medicine and Liver Disease, Strasbourg, France). BHK-21 cell lines were a kind gift from Maria Carla Saleh (Institut Pasteur, Paris, France). Tb1Lu cells were maintained in equal volumes of Ham F-12 (Biowest, Avantor Sciences) and Iscove modified Dulbecco medium (Gibco, Life Technologies) supplemented with 10% fetal bovine serum (FBS, bioSera FB1090/500, Batch# S00O0) in a humidified atmosphere of 5% $CO_2$ at 37°C. All other cells were maintained in Dulbecco's modified Eagle medium (DMEM, Gibco, Life Technologies) supplemented with 10% FBS in a humidified atmosphere of 5% $CO_2$ at 37°C. Tb1Lu, HEK293T and Vero E6 cells are female. MmNE, Huh7 and A549 cells are male. BHK-21 are reported unsexed by ATCC.

WT SINV (strain AR339) and SINV-GFP (kindly provided by Maria Carla Saleh, Institut Pasteur, Paris, France) viral stocks were produced and amplified in BHK21 cells. SFV (strain UVE/SFV/UNK/XX1745; EVAg 001V-02468) was propagated in Vero E6 cells from the initial stock. WT VSV (strain Indiana isolate PI10) and VSV-GFP [35] were propagated in BHK21 cells from the initial stock. Stock titrations were performed by plaque assay on VERO E6 cells. Cells were infected at MOIs ranging from 0.02 to 20, for 8h or 24h, as indicated in the figure legends.

## Plasmids and cloning

pLenti6 Flag-HA-V5 vector was modified from pLenti6-V5 gateway vector (Thermo Fisher scientific V49610) by Gibson cloning (New England Biolabs) as previously described [32], then further modified to remove the V5 tag and ccdB cassette. plenti6 Flag-HA hDicer was already described [32]. mmDicer was cloned from *M. myotis* nasal epithelial cells mRNA using Superscript IV one-step RT-PCR (Thermo Fisher Scientific) with primers Forward (CAGTGTGGTGGAAT-TCTGCAGAAAAGCCCTGCTTTGCAACCC) and Reverse (CCAAACTCATTACTAACCGGTTCAGCTGCTGGGAACCT-GAG). tbDicer was cloned from *T. brasiliensis* lung Tb1Lu cells mRNA using Superscript IV one-step RT-PCR (Thermo Fisher Scientific) with primers Forward (CAGTGTGGTGGAATTCTGCAGAAAAGCCCTGCTTTGCAACCC) and Reverse (CCAAACTCATTACTAACCGGTTCAGCTGTT GGGAACCTGAGG). plenti6 Flag-HA-V5 was digested with AgeI and PstI restriction enzymes, then ligated with amplified mmDicer with matching overhangs by Gibson cloning (New England Biolabs). Catalytic mutations (E1562A and E1819A for mmDicer, E1559A and E1808A for tbDicer) were introduced by PCR inside primers Forward (GACTGCGTGGCTGCCCTGCTGGGCTGCTATTTAA) + Reverse (AGCAGGGCAGCCACG-CAGTCGGCGATGCTTTTGT) and Forward (GGGATATTTTTGCTTCACTTGCTGGTGCCATTTAC) + Reverse (GCAAGT-GAAGCAAAAATATCCCCCATGGCCTTG), then ligated to AgeI and PstI digested plenti6 Flag-HA-V5 by Gibson cloning (New England Biolabs).

## Lentivirus production and generation of stable cell lines

HEK293T cells were plated in 6 well plates and transfected with 0.33 µg pVSV envelope plasmid (Addgene #8454) and 1.33 µg pSPAX2 packaging plasmid (Addgene #12260), and 1.7 µg of either plenti6 Flag-HA Empty, plenti6 Flag-HA hDicer, plenti6 Flag-HA mmDicer, plenti6 Flag-HA mmDicer CM, plenti6 Flag-HA tbDicer or plenti6 Flag-HA tbDicer CM, using Lipofectamine 2000 reagent (Invitrogen, Fisher Scientific) according to the manufacturer's protocol. Media was replaced 6h later, then the supernatant containing viral particles was collected at 48h and filtered through a 0.45 µm PES filter. HEK293T NoDice [38] were then transduced with the filtered lentiviruses mixed with 4µg/ml polybrene. Efficiently transduced cells were selected for several days with blasticidine at 15 µg/mL and polyclonal populations were used for experiments.

## Dicer knock down

*M. myotis* and *T. brasiliensis* Dicer siRNAs were designed in the Dicer 3'UTRs using a service provided by Sigma-Aldrich, Merck (siDicer #1 target sequence: ACCACAAGAGTGCTTTAAT, siDicer #2 target sequence: ACAAGAGTGCTTTAAT-GAT) and ordered through the Dharmacon custom design siRNA tool (Horizon, Revvity). Non-targeting siRNAs with the same chemistry were used as control (Dharmacon, Horizon, Revvity). MmNE or Tb1Lu cells were reverse transfected with siNeg or Dicer siRNAs at 40nM final siRNA concentration using Lipofectamine RNAiMax (ThermoFisher) following manufacturer's protocol. The next day, cells were transfected again with the same siRNAs at 40nM final siRNA concentration, and infected the following day. *M. myotis* Dicer specific siRNAs were designed by siTOOLs Biotech to form a siPool of 30 siRNAs. A siPool of 30 non-targeting siRNAs was used as control (siTOOLs Biotech). MmNE cells were reverse transfected with siPool Neg or siPool Dicer at 3nM final siRNA concentration using Lipofectamine RNAiMax (ThermoFisher) following siTOOLs Biotech instructions. Cells were transfected again the following day with siNeg or siDicer at 3nM final siRNA concentration. The next day, cells were collected to check for efficient knock down or infected with different viruses.

### Live cell imaging

MmNE cells were treated with two rounds of siRNAs then infected with SINV-GFP at MOI 0.2. Cells were then placed into a CellCyte X live cell imaging system (Cytena) where GFP fluorescence and phase contrast pictures (10X objective) of every well were taken every 3h for 72h. The percentage of GFP+ area over the total cell area was quantified with the CellCyte Studio software and averaged from six photos of 2 individual wells per condition.

### Time course infections

For time course viral infections, MmNE cells transfected with two rounds of siRNAs or HEK293T NoDice cells expressing different Dicer constructs were infected with SINV-GFP or VSV-GFP at the indicated MOIs. 1h post infection, the inoculum was removed and replaced with fresh media. Supernatants were collected at this time, as well as at 8h, 24h and 32h post infection to quantify viral titers over time.

### Protein extraction and western blot

Proteins were harvested and homogenized in appropriate quantity of ice-cold lysis buffer (50 mM Tris-HCl pH 7.5, 150 mM NaCl, 5 mM EDTA, 1% Triton X-100, 0.5% SDS protease free inhibitor tablet (complete Mini; Sigma Aldrich)). Protein amounts were quantified using Bradford reagent (Bio-Rad), diluted in lysis buffer + Laemmli and boiled for 5 min before loading. Proteins were loaded on 10% acrylamide-bis-acrylamide gels, or 4–20% Mini-PROTEAN TGX Precast Gels (Bio-Rad) for the Dicer immunoprecipitation experiment. Proteins were separated by migration at 140 V in 1X Tris-Glycine-SDS buffer (Euromedex), then electro-transferred on a nitrocellulose membrane (Amersham 0.45 μm) in 1X Tris-Glycine buffer supplemented with 20% ethanol. Appropriate loading and transfer efficiency were verified by Ponceau S staining (Merck). Membranes were blocked in 5% milk diluted in PBS-Tween 0.2%. Membranes were stained with primary antibodies overnight at 4°C at the following dilutions: anti-DICER (1:2000, A301-937A, Euromedex, Bethyl), anti-HA-HRP (1:1000, 12013819001 Merck, Sigma Aldrich), anti-GFP (1:1000, 11814460001 Merck, Sigma Aldrich), anti-PKR (1:1000, ab32506 Abcam), anti-PACT (1:500, ab75749 Abcam), anti-TRBP (1:500, sc-514124, Cliniscience, Santa Cruz), anti-ADAR1 (1:1000, E6X9R #81284 Cell Signaling), anti-α-Tubulin-HRP (1:10000, Sigma, T6557). After several washes with PBS-Tween 0.2%, the membranes were stained with the following secondary antibodies coupled to horseradish peroxidase (HRP): anti-mouse-HRP (1:4000, A4416 Merck, Sigma-Aldrich) or anti-rabbit-HRP (1:10000, NA9340V Cytiva, Sigma-Aldrich). Detection was carried out using SuperSignal West Femto maximum sensitivity substrate (Pierce, Fisher Scientific) and visualized with a Fusion FX imaging system (Vilber).

### Plaque assay

For SINV-GFP infections, Vero E6 cells were seeded in 96-well plates and infected with 10-fold serial dilutions of infection supernatants. After 1 hour, the inoculum was removed and cells were covered with 2.5% carboxymethyl cellulose and cultured for 72 hours at 37°C in a humidified atmosphere of 5% CO2. Plaques were counted manually under the microscope. For SFV and VSV-GFP infections, Vero E6 cells were seeded in 12-well plates and infected with 10-fold serial dilutions of infection supernatants. After 1 hour, the inoculum was removed, cells were covered with 1.2% Avicel overlay (vivapur mcg, JRS Pharma) and cultured for 48 hours at 37°C in a humidified atmosphere with 5% CO2. Cells were then washed and fixed with 4% formaldehyde (Merck, Sigma Aldrich) in PBS (phosphate buffered saline, Gibco) for 20 minutes, then stained with crystal violet for 5 minutes (Sigma-Aldrich). Plates were washed with water and plaques were counted manually. Viral titers were calculated according to the formula: PFU/mL = #plaques/ (Dilution*Volume of inoculum).

### RNA and small RNA extractions

Total RNA was extracted from cell lysates using Trizol Reagent (15586018, InVitrogen) to which 200 μl of chloroform was added before centrifugation for 15 minutes at 12,000 xg at 4°C. For regular RNA extraction, the aqueous layer was

collected and nucleic acids were precipitated in an equal volume of isopropanol for 10 min at room temperature. For small RNA extraction, the aqueous layer was collected and nucleic acids were precipitated in 3 volumes of 100% ethanol + 10% 1M sodium acetate for 30 minutes at -80°C. In both cases, precipitated RNA was then pelleted for 10 min at 12,000 xg at 4°C then washed with 70% ethanol and reconstituted in RNase free water.

## Northern blot analysis

3 μg of total RNA was loaded onto a 17% urea acrylamide gel and electro separated in a 1X Tris-Borate-EDTA solution. The RNA was then electro transferred to a Hybond-NX membrane (GE Healthcare) and fixed with UV light. Membranes were then pre-hybridized for 30 min with PerfectHyb plus (Sigma Aldrich) at 50°C. Probes consisting of an oligodeoxyri-bonucleotide sequence for which the 5′- was labelled with 25 μCi of [γ-32P]dATP using T4 polynucleotide kinase (Thermo Fisher Scientific) were incubated overnight with the membranes at 50°C. The membranes were then washed twice at 50°C for 10 min with 5X SSC/0.1% SDS, then for 5 min with 1X SSC/0.1% SDS. Finally, membranes were placed in contact with a phosphorimager plate overnight and imaged with a Bioimager FLA-7000 scanner (Fuji). Probe sequences: miR-16 (CGCCAATATTTACGTGCTGCTA), U6 (GCAGGGGCCATGCTAATCTTCTCTGTATCG).

## RT-qPCR analysis

After RNA extraction, samples were incubated with DNase I (Invitrogen, Fisher Scientific) for 20 min at 37°C. The RNA was then reverse-transcribed into cDNA using SuperScript IV (SSIV) reverse transcriptase (Invitrogen, Fisher Scientific) or AMV reverse transcriptase (Promega) with a random nonameric primer, following the manufacturer's protocol. Quantitative real-time PCR was performed using SYBR Green PCR Master Mix (Fisher Scientific) and the following primers: 18S F (CTCTTAGCTGAGTGTCCCGC), 18S R (CTTAATCATGGCCTCAGTTCCGA), SINV F (CCACTACGCAAGCAGAGACG), SINV R (AGTGCCCAGGGCCTGTGTCCG), SFV F (GCGGCAAGAAGGAGAACTGC), SFV R (CAAGCGAAAGCCTC-GTCCAC), VSV F (TGTACAATCAGGCGAGAGTC), VSV R (GGAACTCATGATGAATGGATTGGG).

## Small RNA sequencing and bioinformatic analysis

After small RNA extraction, samples were run on a Bioanalyzer chip to confirm RNA integrity. Samples were then shipped to BGI for library preparation with unique molecular identifiers, and sequenced as single-end 50 base reads on their DNBSEQ sequencing technology platform with at least 20 million reads per sample. Raw reads were pre-cleaned by the sequencing platform, including adapter, contaminant and low-quality removal (Phred+33). Before alignment, redundant sequences were collapsed to one copy using fastq_collapser from FASTX-Toolkit 0.0.14 (http://hannonlab.cshl.edu/fastx_toolkit/) and a read-size filter was applied (18–35 bp). HEK293T and MmNE short reads were then aligned to the human genome assembly GRCh38 (GCF_000001405.40) and the *M. myotis* genome mMyoMyo1.p (GCF_014108235.1), respectively. Mapping was performed using Bowtie 1.3.1 [53] with parameters '-v 1 -a --best --strata'. Unmapped reads were next separately aligned against virus genomes SINV-GFP (private sequence derived from NC_001547.1 – RefSeq database) and VSV-GFP (FJ478454.1) using the following parameters '-v 2 -y -q -m 30 -a --best --strata'. Finally, siRNA signature was examined on 22nt mapped reads by the python tool signature by Antoniewski [37] with options '22 22 1 22'. Sequencing data have been deposited on NCBI's Gene Expression Omnibus [54] and are accessible through GEO Series accession number GSE301457.

## Immunostaining

Cells were seeded onto Labtek eight-chamber slides (Nunc Lab Tek Chamber Slide System) at a density of 8,000 cells per well. The next day, cells were infected for 8h or 24h with WT SINV, WT VSV or SFV. Cells were then fixed for 10 minutes at room temperature with 4% formaldehyde (Sigma-Aldrich) diluted in 1X PBS. Blocking was carried out with 5%

normal goat serum in 0.1% Triton X-100 in 1X PBS for 1 hour at room temperature. Primary antibody staining was carried out for 3h at room temperature using the following antibodies diluted in blocking buffer: anti-Dicer (1:1000, A301-937A, polyclonal, Bethyl) and J2 anti-dsRNA (1:500, RNT-SCI-10010200, Jena Bioscience). The cells were then incubated for 1 hour at room temperature with secondary antibodies goat anti-mouse Alexa 594 (A11032, Invitrogen) or goat anti-rabbit Alexa 488 (A11008, Invitrogen), both diluted at 1:1000 in 1X PBS with 0.1% Triton X-100. Between each step, cells were washed thoroughly with 1X PBS containing 0.1% Triton X-100. Nuclei were stained with DAPI (4',6-diamidino-2-phenylindole, dichlorhydrate, D1306, Thermo Fisher) at 1:5000 in 1X PBS for 5 minutes. Finally, the slides were mounted with coverslips using Fluoromount-G mounting media (Southern Biotech) and observed under a confocal microscope (LSM780, Zeiss).

## Sequential immunostaining and FISH

Mock- or WT SINV-infected cells were cultured on 18 mm round coverslips placed in 12-well plates. Cells were fixed with 3.7% formaldehyde (Sigma-Aldrich) diluted in 1X PBS for 10 minutes at room temperature. Following fixation, cells were permeabilized with 0.1% Triton X-100 in 1X PBS for 5 minutes at room temperature. Immunostaining was performed as described in the previous section but with 1h incubation steps for primary and secondary antibody staining. Subsequently, cells were fixed again with 3.7% formaldehyde (Sigma-Aldrich) diluted in 1X PBS for 10 minutes at room temperature. For RNA detection, cells were incubated overnight at room temperature with SINV genome-specific Stellaris Cal Fluor Red 610 RNA FISH Probes (LGC Biosearch Technologies, custom made by Erika Girardi [55]), kindly provided by Erika Girardi (IBMC, Strasbourg, France). Probes were diluted in RNA FISH hybridization buffer (Stellaris, Biosearch Technologies). Nuclei were stained with DAPI (D1306, Invitrogen, Thermo Fisher Scientific) for 30 minutes. Finally, coverslips were mounted using Fluoromount-G mounting medium (Invitrogen, Thermo Fisher Scientific), and the samples were imaged by fluorescence microscopy (Olympus BX51) or confocal microscopy (LSM780, Zeiss).

## Image analysis

**Colocalization analysis.** All colocalization image analyses were performed on confocal images and using the Fiji software [56]. Detection thresholds and all parameters were standardized between the different images and cell lines to ensure consistency in the analysis. Each analysis was performed on two (SFV) or three (SINV) biological replicates, with each field containing approximately the same number of cells. The "Plot profile" tool on Fiji was used to visualize fluorescent intensity as a function of distance for each channel. The selection of the line or field of observation was made arbitrarily to best represent the whole slide. Threshold parameters were adjusted accordingly to measure the size of various objects, and the minimum size accepted was set to 0.1 µm² to limit background detection. The "Analyse particle" tool was used to extract numerical data, which was then systematically analyzed. The colocalization analysis between two fluorescent signals was performed using the "Colocalization Finder" tool in Fiji. This tool analyzes colocalization on a pixel-by-pixel basis by comparing the intensity of each pixel in one channel to the corresponding pixel in the second channel of a bi-color image. Importantly, the thresholds or brightness levels in the image were not changed during this process. The analysis generates a scatter plot, from which a Pearson correlation coefficient (R) is calculated. A value of +1 indicates perfect correlation, 0 indicates no correlation, and -1 signifies perfect anti-correlation.

**HEK293T NoDice and Dicer knock down analyses.** Images were acquired by epifluorescence microscopy (Olympus BX51), using the same acquisition parameters for all conditions and biological replicates. The light intensity of the lamp was adjusted so that the fluorescence signal would be at the detection limit for MmNE siDicer or HEK293T NoDice EV cells. All images were then taken with the same exposure time and microscope settings. Signal intensity analysis was performed using the Fiji software. For each image, a 70-pixel square was drawn and moved to five different cells picked at random, where the average fluorescence intensity was measured using the « Measure » tool. This process was repeated

on three different images per condition and the resulting measurements were averaged and plotted as one biological replicate. Three biological replicates were quantified overall and plotted as fold change compared to MmNE siNeg or HEK293T NoDice EV cells.

## Dicer immunoprecipitation

NoDice cells expressing different Flag-HA tagged Dicer constructs were plated and mock-infected or infected with SINV-GFP at MOI 0.02. 24 hours later, cells were harvested, washed twice with ice-cold 1X PBS (Gibco, Life Technologies), and resuspended in lysis buffer (50 mM Tris-HCl pH 7.5, 140 mM NaCl, 6 mM MgCl2, 0.1% NP-40), supplemented with a Complete-EDTA-free Protease Inhibitor Cocktail tablet (complete Mini; Sigma Aldrich). Cells were lysed on ice for 10 min and debris were removed by 10 min centrifugation at 12,000 $xg$ at 4°C. An aliquot of the cleared lysates was kept aside as protein input. The rest of the samples were divided in two equal volumes and incubated with 40 µL magnetic microparticles coated with monoclonal anti-HA or anti-MYC antibodies (MACS purification system, Miltenyi Biotech) at 4°C for 1 hour with constant rotation. Next, samples were passed through µ Columns (MACS purification system, Miltenyi Biotech) and washed four times with 200 µL lysis buffer. Finally, proteins were eluted with 2X Laemmli buffer (10% glycerol, 4% SDS, 62.5 mM TrisHCl pH 6.8, 5% (v/v) 2-β-mercaptoethanol, Bromophenol Blue) that was pre-warmed to 95°C.

## Comparison of Dicer protein sequences

Dicer protein sequences from different bat species studied in the literature and belonging to various families were extracted from the UCSC Genome Browser – Bat1K Genomes database [57]. The amino acid sequences were aligned using the « Align » tool available on the UniProt platform, and the percentage of homology between each protein sequences was calculated.

## Supporting information

**S1 Fig. Small RNAseq results for the second replicate of SINV-GFP infected HEK293T and *M. myotis* nasal epithelial cells.** HEK293T or MmNE cells were mock infected or infected with SINV-GFP, then small RNAs were extracted and sequenced. n = 2 independent experiments, see Fig 1 for data from the other replicate. (A) Number of reads that align to the SINV-GFP genome for each cell type based on the read size. The total number of reads is further broken down into colors based on the identity of the first nucleotide of the read: yellow for U, blue for A, purple for G and black for C. (B) Location of the 22 nucleotide (nt) reads along the SINV-GFP genome for each sample. The number of reads falling on the same nucleotide is represented in red if the reads align to the genome (+ strand) and in blue if they align to the anti-genome (- strand). (C) For each sample, 22 nt small RNA pairs that overlap on + and - strands were analyzed and associated z-scores [37] were plotted for the indicated nucleotide overlaps.
(TIFF)

**S2 Fig. Small RNAseq results for VSV-GFP infected HEK293T and *M. myotis* nasal epithelial cells.** HEK293T or MmNE cells were mock infected or infected with VSV-GFP, then small RNAs were extracted and sequenced. n = 2 independent experiments. (A) Number of reads that align to the VSV-GFP genome for each cell type based on the read size. The total number of reads is further broken down into colors based on the identity of the first nucleotide of the read: yellow for U, blue for A, purple for G and black for C. (B) Location of the 22 nucleotide (nt) reads along the VSV-GFP genome for each sample. The number of reads falling on the same nucleotide is represented in red if the reads align to the genome (+ strand) and in blue if they align to the anti-genome (- strand). For each sample, the number of 22 nt small RNA pairs that overlap on + and - strands was so small (2 and 0 for the HEK293T replicates, 6 and 29 for the MmNE replicates) that the associated Z scores [37] were not plotted.
(TIFF)

**S3 Fig. Dicer does not protect against SINV in *M. myotis* nasal epithelial cells.** (A-C) MmNE cells treated with two different siRNAs targeting Dicer (siDicer) or a non-targeting control (siNeg) were infected with SINV-GFP for 24h at MOI 0.2. (A) Protein lysates were collected and analyzed by western blot using antibodies for Dicer and GFP. α-Tubulin was used as loading control. Images are representative of 3 independent experiments. GFP band intensities relative to Tubulin band intensities were quantified with ImageJ for each experiment, and means+standard deviations were plotted relative to siNeg infected samples. (B) RNAs were purified and levels of SINV-GFP RNA were quantified by qRT-PCR. Means+standard deviations were plotted relative to siNeg infected samples for 3 independent experiments. (C) Supernatants were collected and viral titers were quantified by plaque assay. Means+standard deviations were plotted for 3 independent experiments. For the RNA and protein graphs, p values were calculated using one sample t tests compared to 1. For the viral titer graph, p values were calculated using an ordinary one-way ANOVA with Dunnett's multiple comparison test with a single pooled variance compared to siNeg. * p<0.05; ns: not significant. (D) MmNE cells treated with siPool siRNAs targeting Dicer (siPool Dicer) or a non-targeting control (siPool Neg) were infected with SINV-GFP at MOI 0.2 (left) or 2 (right) for 1h, then the inoculum was removed and replaced with fresh media. Supernatants were collected at this time and at 8h, 24h and 32h post infection. Viral titers were quantified by plaque assay. Means+standard deviations were plotted for 3 independent experiments. (TIFF)

**S4 Fig. Human Dicer antibodies can be used for immunofluorescence staining of mmDicer.** (A-B) MmNE cells treated with siPool siRNAs targeting Dicer (siPool Dicer) or a non-targeting control (siPool Neg) were fixed 48 hours after the last transfection. (A) Dicer staining (yellow) was imaged by epifluorescence microscopy. DAPI staining (blue) indicates cell nuclei. Images are representative of 4 independent experiments. Scale bars 10µm. (B) For each experiment, Dicer staining intensity was quantified and averaged for 3 photos per condition, then normalized to the siPool Neg condition. Means+standard deviations were plotted for all 4 independent experiments. (C-D) HEK293T NoDice cells expressing Flag-HA tagged hDicer, mmDicer or an empty vector control were stained for Dicer (yellow) and Flag (magenta) and imaged by epifluorescence microscopy. DAPI staining (blue) indicates cell nuclei. Images are representative of 3 independent experiments. Scale bars 10µm. Co-localization between Dicer and Flag signals was quantified for the representative images and resulting Pearson correlation R coefficients are indicated on the right. (D) For each experiment, Dicer staining intensity was quantified and averaged for 3 photos per condition, then normalized to the NoDice EV condition. Means+standard deviations were plotted for all 3 independent experiments. For all graphs, p values were calculated using one sample t tests compared to 1. * p<0.05; ns: not significant. (TIFF)

**S5 Fig. Dicer foci are not observed in SINV-infected human cell lines or in VSV infected cells.** (A) Human A549 and Huh7 cells were mock infected or infected with WT SINV at 24h at MOI 0.2 and 0.02, respectively. Localization of Dicer (yellow) and dsRNA via J2 antibodies (magenta) were imaged by immunofluorescence and confocal microscopy. DAPI staining (blue) indicates cell nuclei. Images are representative of 2 independent experiments. Scale bars 10µm. (B) HEK293T or MmNE cells were infected with WT VSV for 8h at MOI 1. Localization of Dicer (yellow) and dsRNA via J2 antibodies (magenta) were imaged by immunofluorescence and confocal microscopy. DAPI staining (blue) indicates cell nuclei. Images are representative of 2 independent experiments. Scale bars 10µm. (C) HEK293T or MmNE cells were infected with WT SINV for 24h at MOI 0.02 or 4, respectively. Localization of Dicer (yellow) by immunofluorescence and of SINV plus strand (+) RNA (magenta) by FISH was imaged by confocal microscopy. See Fig 4C for epifluorescence microscopy images of the same samples. DAPI staining (blue) indicates cell nuclei. Images are representative of 2 independent experiments. Scale bars 10µm. A zoomed-in image of the boxed area in the merge images was added. The graphs show the signals for each channel along the line drawn in the zoomed in images. The staining intensity of SINV plus strand (+) RNA was too low on confocal images to calculate accurate Pearson correlation R coefficients. (TIFF)

**S6 Fig. *M. myotis* Dicer expressed in human NoDice cells protects from SINV-GFP but not VSV-GFP infection.**
(A) Expression of Dicer was analyzed by western blot for MmNE cells or HEK293T NoDice cells expressing Flag-HA-tagged mmDicer or an empty vector control. α-Tubulin was used as loading control. (B) HEK293T NoDice cells expressing Flag-HA-tagged hDicer or mmDicer were mock infected or infected with SINV-GFP for 24h at MOI 0.02. Cells were then lysed, a small aliquot was kept as input, and Dicer was immunoprecipitated with anti-HA coated magnetic beads (IP Dicer) or anti-myc coated beads (IP ctrl). Eluted proteins were analyzed by western blot using antibodies for the known hDicer interactors ADAR1, PKR, TRBP and PACT. Anti-HA antibodies show efficient immunoprecipitation of hDicer and mmDicer. Images representative of 2 independent experiments. (C-D) HEK293T NoDice cells expressing Flag-HA-tagged hDicer, mmDicer or an empty vector control were infected with (C) VSV-GFP at MOI 0.02 or (D) SINV-GFP at MOI 0.002 (left) or 0.02 (right) for 1h, then the inoculum was removed and replaced with fresh media. Supernatants were collected at this time and at 8h, 24h and 32h post infection. Viral titers were quantified by plaque assay. Means + standard deviations were plotted for 3 independent experiments. (E-G) HEK293T NoDice cells expressing Flag-HA tagged hDicer, mmDicer, mmDicer CM, tbDicer, tbDicer CM or an empty vector control were infected with VSV-GFP at MOI 0.02 for 24h. (E) Protein lysates were collected and analyzed by western blot using antibodies for GFP. α-Tubulin was used as loading control. Images are representative of 3 (mmDicer CM) or 4 independent experiments. GFP band intensities relative to Tubulin band intensities were quantified with ImageJ and the mean + standard deviation was plotted relative to empty vector control for each experiment. (F) RNAs were purified and levels of VSV-GFP RNA were quantified by qRT-PCR. Means + standard deviation normalized to 18S were plotted relative to empty vector control for 3 (mmDicer CM) or 4 independent experiments. (G) Supernatants were collected and viral titers were quantified by plaque assay. Means + standard deviations were plotted for 3 (mmDicer CM) or 4 independent experiments. Each symbol represents a separate experiment. For the RNA and protein graphs, p values were calculated using one sample t tests compared to 1. For the viral titer graph, p values were calculated using an ordinary one-way ANOVA with Dunnett's multiple comparison test with a single pooled variance compared to EV. ns: not significant.
(TIFF)

**S7 Fig. Dicer amino acid sequence is highly conserved across different bat species and with human Dicer.** Percentage homology was calculated for each Dicer amino acid sequence from the indicated species and represented as a similarity matrix.
(TIFF)

## Acknowledgments

We would like to thank members of the laboratory, as well as Lucie Etienne (CIRI, Lyon, France) and Nolwenn Jouvenet (Institut Pasteur, Paris, France) for insightful discussions, suggestions and for sharing material.

## Author contributions

**Conceptualization:** Léa Gaucherand, Sébastien Pfeffer.

**Data curation:** Julie Cremaschi.

**Formal analysis:** Léa Gaucherand, Hugo Marie.

**Funding acquisition:** Sébastien Pfeffer.

**Investigation:** Léa Gaucherand, Hugo Marie.

**Methodology:** Léa Gaucherand.

**Project administration:** Sébastien Pfeffer.

**Software:** Julie Cremaschi.

**Supervision:** Léa Gaucherand, Sébastien Pfeffer.

**Visualization:** Hugo Marie, Julie Cremaschi.

**Writing – original draft:** Léa Gaucherand.

**Writing – review & editing:** Hugo Marie, Julie Cremaschi, Sébastien Pfeffer.

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
