## [Editor Report · Decision Letter 0]

26 Sep 2025

Dicer accumulates in cytoplasmic foci upon alphavirus infection and plays a proviral role in Myotis myotis bat cells

PLOS Pathogens

Dear Dr. Pfeffer,

Thank you for submitting your manuscript to PLOS Pathogens. After careful consideration, we feel that it has merit but does not fully meet PLOS Pathogens's publication criteria as it currently stands. Therefore, we invite you to submit a revised version of the manuscript that addresses the points raised during the review process at Review Commons, according to the submitted Revision Plan (see below for details).

Please submit your revised manuscript within 60 days Nov 25 2025 11:59PM. If you will need more time than this to complete your revisions, please reply to this message or contact the journal office at plospathogens@plos.org. Please include the following items when submitting your revised manuscript:

We look forward to receiving your revised manuscript.

Kind regards,

Ronald P van Rij

Guest Editor

PLOS Pathogens

Alexander Gorbalenya

Section Editor

PLOS Pathogens

Editor-in-Chief

PLOS Pathogens

orcid.org/0000-0003-2946-9497

Editor-in-Chief

PLOS Pathogens

orcid.org/0000-0002-7699-2064

**Additional Editor Comments:**

The plan for revision is solid, and I invite you to submit a revised version of the manuscript addressing the reviewers’ comments as proposed.

Regarding the experiments that the authors prefer to not carry out: I agree that the flow cytometry assessment of Dicer expression is not essential if microscopy images are provided. I agree with the reviewers that experiments in other bat cells/with the Dicer protein of other bat species would be a valuable addition to the manuscript. However, if the authors cannot obtain other bat cell lines, the proposed experiments with only Tblu cells / T. brasiliensis Dicer would suffice.

Other suggestion:

I wonder whether the Z score plots of the VSV infections (in HEK and M myotis cells; Fig. 3D) are informative, given that they are based on very few overlapping reads. The authors may consider removing these plots, or alternatively, indicating in the figure the number of pairs underlying the plot.

**Journal Requirements:**

https://journals.plos.org/plospathogens/s/submission-guidelines#loc-parts-of-a-submission

4) We notice that your supplementary Figures are included in the manuscript file. Please remove them and upload them with the file type 'Supporting Information'. Please ensure that each Supporting Information file has a legend listed in the manuscript after the references list.

5) Please ensure that the funders and grant numbers match between the Financial Disclosure field and the Funding Information tab in your submission form. Note that the funders must be provided in the same order in both places as well.

**Reviewers' Comments:**

**Figure resubmission:**

**Reproducibility:**



---

## [Editor Report · Decision Letter 1]

3 Dec 2025

PPATHOGENS-D-25-02282R1

Bat Dicer antiviral role and subcellular localization differ upon alphavirus infection in two distinct species

PLOS Pathogens

Dear Dr. Pfeffer,

Thank you for submitting your manuscript to PLOS Pathogens. After careful consideration, we feel that the manuscript has greatly improved but does not yet fully meet PLOS Pathogens's publication criteria as it currently stands. Therefore, we invite you to submit a revised version of the manuscript that addresses the final point raised by the editor.

We look forward to receiving your revised manuscript.

Kind regards,

Ronald P van Rij

Guest Editor

PLOS Pathogens

Alexander Gorbalenya

Section Editor

PLOS Pathogens

Sumita Bhaduri-McIntosh

Editor-in-Chief

PLOS Pathogens

orcid.org/0000-0003-2946-9497

Michael Malim

Editor-in-Chief

PLOS Pathogens

orcid.org/0000-0002-7699-2064

**Editor Comments :**

The authors have done extensive additional work to address the review comments (Review Commons) and restructured the manuscript. The conclusions are well supported by the data and the manuscript provides interesting new insights in the role of RNAi in bat cells.

There is still one experiment that I consider essential before the manuscript can be accepted:

- (viral) small RNA profiling of infected Tb1Luc cells. The authors suggest that Dicer has antiviral activity in Tadarida brasiliensis Tb1Lu cells, which contrasts with their observation in Myotis myotis cells. It will be important to study whether these cells produce viral siRNAs as an indication that the observed antiviral effect is small RNA/RNAi dependent.

**Journal Requirements:**

1)  Thank you for stating "Sequencing data have been deposited on NCBI’s Gene Expression Omnibus (Edgar et al, 2002) and are accessible through GEO Series accession number GSE301457." Please note that, though access restrictions are acceptable now, your entire minimal dataset will need to be made freely accessible if your manuscript is accepted for publication. This policy applies to all data except where public deposition would breach compliance with the protocol approved by your research ethics board.

**Figure resubmission:**
---

## [Editor Report · Decision Letter 2]

15 Dec 2025

Dear Dr. Pfeffer,

We are pleased to inform you that your manuscript 'Bat Dicer antiviral role and subcellular localization differ upon alphavirus infection in two distinct species' has been provisionally accepted for publication in PLOS Pathogens.

Best regards,

Ronald P van Rij

Guest Editor

PLOS Pathogens

Alexander Gorbalenya

Section Editor

PLOS Pathogens

Sumita Bhaduri-McIntosh

Editor-in-Chief

PLOS Pathogens

orcid.org/0000-0003-2946-9497

Michael Malim

Editor-in-Chief

PLOS Pathogens

orcid.org/0000-0002-7699-2064

The authors have clearly explained (in rebuttal and discussion of the manuscript) their rationale for not investing time and resources into siRNA profiling of the Tadarida brasiliensis cells. While small RNA profiles would still be of significant interest, I agree with the authors that it is not essential to support their main conclusions and I therefore propose to accept the manuscript.
---

## [Editor Report · Acceptance letter]

Dear Dr. Pfeffer,

We are delighted to inform you that your manuscript, "Bat Dicer antiviral role and subcellular localization differ upon alphavirus infection in two distinct species," has been formally accepted for publication in PLOS Pathogens.

Best regards,

Sumita Bhaduri-McIntosh

Editor-in-Chief

PLOS Pathogens

orcid.org/0000-0003-2946-9497

Michael Malim

Editor-in-Chief

PLOS Pathogens

orcid.org/0000-0002-7699-2064